# The pre-anesthetic period is the best time to evaluate the knee flexion angle for predicting the flexion angle after total knee arthroplasty: A prospective cohort study

Pakpoom Ruangsomboon[1,2], Chaturong Pornrattanamaneewong[1], Polasan Santanapipatkul[3], Sorarid Sarirasririd[4], Keerati Chareancholvanich[1], Rapeepat Narkbunnam[1]*

1 Department of Orthopaedic Surgery, Faculty of Medicine, Siriraj Hospital, Mahidol University, Bangkok, Thailand, 2 Institute of Health Policy, Management and Evaluation, University of Toronto, Toronto, Ontario, Canada, 3 Department of Orthopaedic, Samutsakorn Hospital, Samutkakorn, Thailand, 4 Orthopaedic Center, Siriraj Piyamaharajkarun Hospital, Bangkok, Thailand

* mai_parma@hotmail.com

## Abstract

### Introduction

Knee flexion angle (KFA) is one of the most critical factors for evaluating patient functional outcomes after total knee arthroplasty (TKA). Preoperative KFA and intraoperative drop leg test are both accepted as predictors of postoperative KFA after TKA. Preoperative testing performed after anesthesia helps overcome pain-related limitations; however, the KFA measurement timepoint that best predicts KFA at 6 months after TKA has not yet been established.

### Methods

This prospective cohort study recruited patients who underwent unilateral primary TKA at Siriraj Hospital (Bangkok, Thailand) during August 2012 to August 2017. We recorded KFA at the pre-anesthetic phase, post-anesthetic phase, intraoperation using drop leg test, and at 6-months post-operation. Pearson's correlation coefficient was used to evaluate correlation between different measurement timepoints and 6 months after surgery. Those same relationships were evaluated for overall patients, and for patients with KFA <90˚ (poor KFA), 90–120˚ (average KFA), and >120˚ (high KFA).

### Results

A total of 165 patients with a mean age of 68.7 years were recruited. Pre-anesthetic KFA measurement had the highest positive correlation with the 6-month KFA ($r = 0.771$, $p<0.05$). Post-anesthetic measurement and intraoperative drop leg KFA measurement had moderate positive correlation ($r = 0.561$, $p<0.05$) and low positive correlation ($r = 0.368$, $p<0.05$) with the 6-month KFA, respectively. The average KFA group had the highest positive correlation between pre-anesthetic KFA measurement and the 6-month KFA ($r = 0.711$, $p<0.05$).

**Data Availability Statement:** Data are available from the Siriraj Institutional Data Access / Ethics Committee (contact via orthoresearch.si@gmail.com) for researchers who meet the criteria for access to confidential data regarding potentially identifying data-sensitive patient information.

**Funding:** The authors received no specific funding for this work.

**Competing interests:** The authors have declared that no competing interests exist.

Predicted 6-month KFA (degrees) adjusted for pre-anesthetic KFA is 45.378 + [0.596 x pre-anesthetic KFA (degrees)] (r = 0.67, p <0.05).

## Conclusions

Pre-anesthetic KFA demonstrated the highest correlation with the final KFA at six months after unilateral primary TKA, especially in the patients who had a preoperative KFA within 90–120˚.

## Introduction

Total knee arthroplasty (TKA) has been a successful treatment for end-stage knee osteoarthritis for at least 25 years, with a reported 82% survival rate [1]. However, 8–10% of patients remain dissatisfied their TKA within the first 5-years post-TKA [2]. Previous studies reported association between TKA patient dissatisfaction and both worsening knee functional scores and decreased postoperative range of motion (ROM) [3, 4].

Knee flexion angle (KFA) is one of the essential knee motion parameters. Knee flexion activity is particularly important for Asians because their normal daily activities involve a lot of kneeling, squatting, floor sitting, gardening, sexual performances, praying, dining, and using the oriental toilet [5–8]. Thus, predicting long-term KFA after TKA is very important in Asian population. There are generally three different timepoints during the perioperative period in which KFA can be evaluated. The first is the pre-anesthetic phase. The second is the post-anesthetic phase, also known as examination under anesthesia (EUA), to ensure adequate pain control [9]. The third is KFA measurement via the intraoperative drop leg test, which was proposed by Lee, et al. [10]. Post-anesthetic KFA measurement and KFA measurement by way of the intraoperative drop-leg test are the reference values that surgeons have commonly used to advise patients regarding their final KFA. However, many surgeons discovered that these parameters did not always accurately reflect the patient's KFA during follow-up. To the best of our knowledge, no previous study has compared the correlations or the predictive accuracy of KFAs measured at these different perioperative timepoints with the final KFA. Previous evidence have reported that the steady phase of knee ROM recovery after TKA could be reached at six-months post-operation [11–14].

The aim of this study was to identify the perioperative KFA measurement timepoint that most strongly correlates with the KFA at six months after TKA.

## Materials and methods

This prospective cohort study was conducted at the Division of Adult Reconstructive Surgery, Department of Orthopaedic Surgery, Faculty of Medicine Siriraj Hospital, Mahidol University, Bangkok, Thailand. This study was registered at the clinicaltrials.in.th website (reg. no. TCTR 20210202003) and had ethical approval. Informed consent was written from all participants. All patients were recruited from the preoperative arthroplasty clinic of Siriraj Hospital. Patients aged over 55 years with the diagnosis of primary osteoarthritic knee who were scheduled for unilateral primary TKA were included. The exclusion criteria were 1) prior hip and knee replacement surgery on any side; 2) patients requiring metal augmentation, stem, or constrained prosthesis; 3) patients requiring intraoperative patellar resurfacing; 4) intraoperative or postoperative complication requiring immobilization or reoperation of the knee; 5) loss to

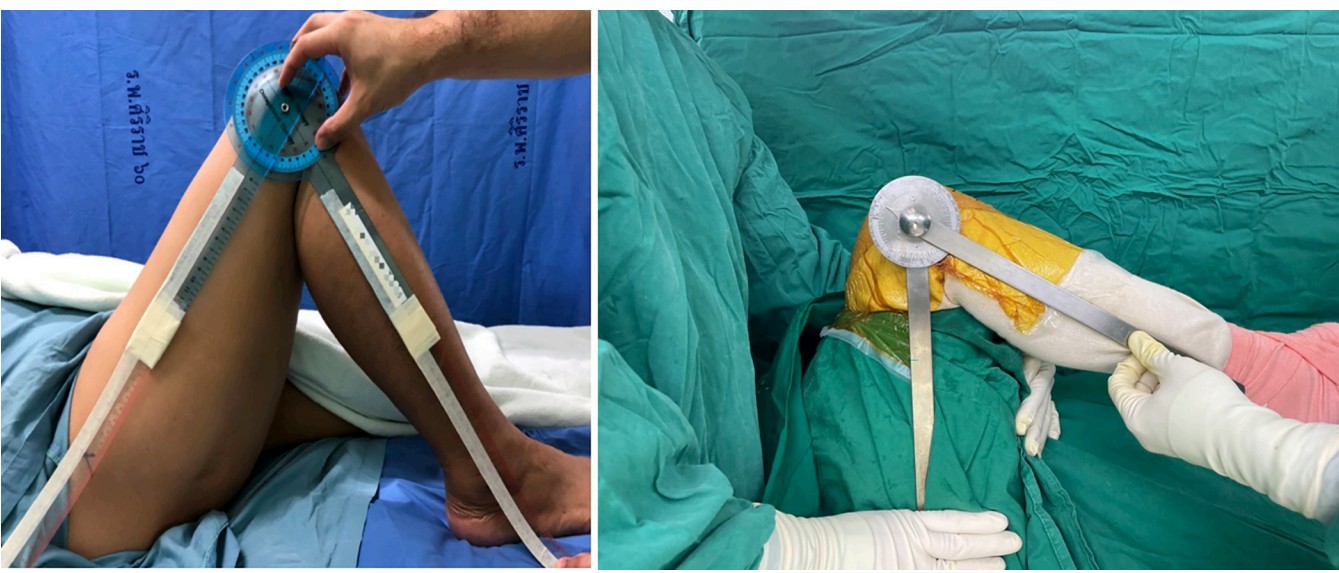

**Fig 1. Knee flexion angle (KFA) measurement using a long protractor double-arm standard goniometer (left).** Intraoperative drop leg test KFA was measured using a long arm sterile protractor after placement of the prosthesis and capsular closure (right).

follow-up before six months after surgery; and/or, 6) stiff knee requiring manipulation under anesthesia (MUA). Because prior hip and opposite knee surgery can alter the post-operative KFA, these conditions were excluded [15]. We also excluded TKA that required MUA because we intended to focus our study question in primary TKA with no stiffness related to the surgery and no abnormal biologic responses that may turn into arthrofibrosis. Such excluded cases may have different recovery courses after TKA that are usually related to surgical issues and/or abnormal biological responses [16].

Before starting the administration of anesthesia in the operating theatre, active KFA was measured and defined as the pre-anesthetic KFA (Fig 1). All patients received the same anesthetic protocol using spinal anesthesia and adductor canal block. The attending anesthesiologist confirmed the anesthetic effect before examination under anesthesia (EUA) was performed and the post-anesthetic KFA was measured. The same examiner, an arthroplasty fellow, performed and recorded all evaluations in the same manner.

Three experienced arthroplasty surgeons performed all TKAs using a similar technique. Tourniquet pressure of 300 mmHg was applied after exsanguinating blood using a rubber band before the skin incision was made. All operations were performed using the mini-medial parapatellar approach, and the measured resection technique. In all cases, surgeons targeted femoral and tibial cuts perpendicular to the mechanical axis according to the preoperative angles as measured by preoperative scanogram [17]. The rotation of the femur was adjusted to achieve a balanced, rectangular flexion gap and proper patellar tracking. To obtain a rectangular flexion gap, we set femoral rotation parallel to or with slightly external rotation to the surgical transepicondylar axis. The distal femoral cut and the proximal tibial cut were performed All patients received cemented NexGen LPS-Flex fixed-bearing prosthesis without patellar resurfacing (Zimmer Biomet, Warsaw, IN, USA). After inserting the femoral and tibial prosthesis and performing capsular closure, the intraoperative drop leg test was performed using a long sterile goniometer. Intraoperative flexion against gravity or intraoperative drop leg test was conducted after capsular closure by passively flexing the patient's hip 90 degrees and allowing the patient's lower leg weight to flex the knee joint [10]. The intraoperative drop leg

KFA was then recorded (Fig 1). We used periarticular analgesic injection with 0.5% bupiva-caine 20 cc, ketorolac 30 mg, and 1% lidocaine with adrenaline 1 ml. At the end of the opera-tion, a vacuum drain was placed intraarticularly that was retained for 48 hours. The tourniquet was deflated after skin closure. Non-compressive dressing was applied in any study patients.

All patients received similar postoperative pain management, including intravenous non-steroidal anti-inflammatory drugs (NSAIDs) for two days followed by oral NSAIDs and para-cetamol with codeine. Intravenous morphine was given as a rescue drug if the numerical rating scale for pain (NRS-pain) score [18] was equal to or greater than four. Patients were mobilized under a physiotherapist's supervision on the first postoperative day onwards. Range-of-motion (ROM) exercise as tolerated was encouraged during the postoperative hospital stay. The con-tinuous passive motion machine was not used during rehabilitation. Before discharge–a home exercise program, including ambulation training, ROM, and strengthening exercise, was pro-vided to all study patients. At six months after surgery, active KFA was measured and defined as the 6-month KFA.

### Measurement of knee flexion angle (KFA)

We used a long-arm goniometer to measure KFA, as suggested by Handcock GE, et al.. This method showed high inter-rater and intra-rater reliability with intraclass correlation coeffi-cients of more than 0.99 and 0.98, respectively [19]. In the present study, measuring KFA at each time point was performed and recorded in the same manner. The recorded KFA was an average of the angle we retrieved after measuring consecutively twice by the measurer, who had been trained and tested for reliability before collecting the data. KFA was measured with patients in the supine position on an operating table. The anatomical landmarks were the tip of the greater trochanter, the lateral femoral epicondyle, and the tip of the lateral malleolus, all of which were palpated and marked. A line between the tip of the greater trochanter and the lateral femoral epicondyle was defined as the femoral axis. A line between the lateral femoral epicondyle and the lateral malleolus was defined as the tibial axis. Using the lateral femoral condyle as a fulcrum, KFA was the angle between the femoral and tibial axis. KFA was mea-sured and recorded at the pre-anesthetic period, the post-anesthetic period, and during the intraoperative period using drop-leg test (Fig 1). KFA at the 6-month follow-up was evaluated in the same manner.

### Sample size calculation

The primary outcome was the correlation between the 6-month KFA and each of the three perioperative KFAs. A previous study by Kotani, et al. [20] found the strength of correlation between the preoperative and postoperative ROM to be weakly positive (r = 0.32). Our power analysis and sample size calculation revealed that a sample of 160 knees would yield 80% statis-tical power to detect a 0.25 effect size between groups (type I error: 0.05, type II error: 0.20). Our decision to increase enrollment by 10% to compensate for loss to follow-up for any reason increased the minimum number of knees to 176.

### Statistical analysis

The data were analyzed using SPSS Statistics v18.0 (SPSS, Inc., Chicago, IL, USA). Quantitative data are presented as mean ± standard deviation (SD), and qualitative data are given as fre-quency and percentage. The normality of data distribution was tested by Kolmogorov-Smir-nov test. Pearson's correlation coefficient was used to evaluate correlation between the perioperative KFA at each timepoint (preoperative, intraoperative, and postoperative) and the 6-month KFA. A correlation coefficient of 0.1–0.3 was considered a weak correlation, a

coefficient of 0.3–0.6 was considered a moderate correlation, and a coefficient of more than 0.6 was considered a strong correlation [21]. To assess whether the strength of correlation differs among different KFA ranges, patient KFA measurements at each timepoint were further categorized into three subgroups, including less than 90˚ (poor KFA), 90–120˚ (average KFA), and more than 120˚ (high KFA). Paired t-tests were performed to compare the pre- and post-anesthetic KFA, and the post-anesthetic and intraoperative drop-leg KFA. A p-value<0.05 was considered to be statistically significant. We employed linear regression analysis to obtain an equation for predicting the final KFA based on the pre-operative KFA.

## Results

A total of 176 patients were recruited. Eleven patients were excluded due to the following reasons: complicated procedure (3 cases), intraoperative femoral condyle fracture (2 cases), stiff knee requiring manipulation under anesthesia (4 cases), and loss to follow-up (2 cases). The remaining 165 patients were included in our final analysis. Kolmogorov-Smirnov test revealed normal distribution in all data. As shown in Table 1, The mean age of participants was 68.8 ±7.9 years, and most patients were female (87.0%). The mean body mass index (BMI), weight, and height was 27.8±4.9 kg/m$^2$, 66.8±12.1 kg, and 155.0±8.0 cm, respectively. There were no extreme outliers over 2.5 standard deviation in our cohort's demographic data. The strength of correlations between the KFA at each perioperative timepoint and the 6-month KFA are shown in Table 2. The pre-anesthetic KFA had the strongest positive correlation with the 6-month KFA (r = 0.771, p<0.05). The post-anesthetic KFA yielded a moderate positive correlation (r = 0.561, p<0.05), and the intraoperative drop leg KFA produced the lowest correlation (r = 0.368, p<0.05) with the 6-month KFA.

We then categorized patients into three groups according to their perioperative KFA. There were 9 (5.5%), 108 (65.4%), and 48 (29.1%) patients in the poor (<90 degrees), average (90–120 degrees), and high (>120 degrees) pre-anesthetic KFA groups, respectively. Baseline characteristics among the subgroups were similar (Table 1). We found that only the average pre-anesthetic KFA group had a strong positive correlation with the 6-month KFA (r = 0.711, p<0.001). The average post-anesthetic KFA group had only a moderate positive correlation with the 6-month KFA (r = 0.408, p<0.001). The poor KFA group could not demonstrate significant correlations with the 6-month KFA at any timepoint. Although there was a moderate negative correlation between the high pre-anesthetic KFA group and the 6-month KFA (r = -0.478, p<0.001), no other significant correlations between the high KFA group and the final KFA were observed.

**Table 1. Patient demographic and anthropometric characteristics.**

| Characteristics | Primary analysis | Post hoc analysis | | |
|---|---|---|---|---|
| | Overall | Group A | Group B | Group C |
| | (N = 165) | (poor KFA) | (average KFA) | (high KFA) |
| | | (n = 9) | (n = 108) | (n = 48) |
| Age (years) | 68.8±7.9 | 70.1±7.6 | 67.9±6.9 | 70.5±9.5 |
| Female to male ratio | 145:20 | 8:1 | 94:14 | 43:5 |
| Body mass index (kg/m$^2$) | 27.8±4.9 | 29.0±5.8 | 28.3±5.2 | 26.2±3.2 |
| Weight (kg) | 66.8±12.1 | 68.5±14.6 | 67.3±12.4 | 65.1±10.7 |
| Height (cm) | 155.0±8.0 | 154.0±13.7 | 154.0±6.8 | 157.3±8.7 |
| Right to left side ratio | 89:76 | 6:3 | 70:38 | 22:26 |

Data presented as mean ± standard deviation or ratio

**Abbreviation:** KFA, knee flexion angle

**Table 2. Correlation between different measurement timepoints and 6 months after surgery stratified by patient group.**

| Patient group | n | Pearson's correlation coefficient | | |
|---|---|---|---|---|
| | | Pre-anesthetic *vs*. 6-month KFA | Post-anesthetic *vs*. 6-month KFA | Intraoperative drop leg test *vs*. 6-month KFA |
| Overall | 165 | 0.771* | 0.561* | 0.368* |
| Group A (poor KFA) | 9 | 0.106 | 0.150 | 0.050 |
| Group B (average KFA) | 108 | 0.711* | 0.408* | 0.145 |
| Group C (high KFA) | 48 | -0.478* | -0.174 | -0.224 |

*$p$-value<0.05

**Abbreviation:** KFA, knee flexion angle

Table 3 illustrates the KFA measured at different timepoints in the study. Among all 165 cases, the mean pre-anesthetic KFA was 112.9 degrees, which is identical to the mean 6-month KFA. The mean post-anesthetic KFA was generally higher than the pre-anesthetic KFA. The mean intraoperative KFA was higher than the preoperative and postoperative KFAs, except in the high KFA subgroup. Moreover, there was a marked increase in the mean KFA from the pre-anesthetic phase to the 6-month postoperative time-point in the poor KFA group. The average KFA group had a slightly increased KFA at 6-months post-operation compared to the pre-anesthetic KFA measurement timepoint. In contrast, the pre-anesthetic KFA in the high KFA group was higher than the 6-month KFA. A formula for predicting 6-month KFA (degrees) from pre-anesthetic KFA is 45.378 + [0.596 x pre-anesthetic KFA (degrees)] in over-all patient (r = 0.67, p<0.05).

## Discussion

The most important finding of the study was that the correlation between the pre-anesthetic KFA and the 6-month KFA was the highest. The post-anesthetic KFA and intraoperative drop leg test KFA might not be appropriate predictors of 6-month KFA because they yielded insignificant and negligible correlations. This is an important finding because KFA is utilized in clinical practice to inform patients about their predicted postoperative KFA. If the given KFA prediction is inaccurate, TKA patients may be dissatisfied if their postoperative KFA does not meet their expectations. As a result, attempts have been made to create a system that can accurately predict postoperative KFA in TKA patients. This field of research had been done in a variety of ways, including focusing on the contralateral KFA to predict the final KFA [15]. Previous studies also found preoperative KFA to be a good predictor of the 6-month KFA [22–25]. However, it was proposed that many end-stage osteoarthritis knee patients experience pain, which might have limited their preoperative KFA results [26]. This dilemma resulted in a

**Table 3. Mean knee flexion angle (KFA) at each timepoint stratified by patient group.**

| Patient group | n | Degrees (mean±SD) | | | | |
|---|---|---|---|---|---|---|
| | | Pre-anesthetic KFA | Post-anesthetic KFA | Intraoperative Drop leg test KFA | 6-month KFA | Change of pre-anesthetic vs 6-month KFA |
| Overall | 165 | 112.9±17.3 | 122.8±13.9 | 124.7±10.5 | 112.9±13.0 | 0±1.7 |
| Group A (poor KFA) | 9 | 77.8±5.1 | 102.2±16.6 | 109.4±15.5 | 92.8±10.3 | 15±3.8 |
| Group B (average KFA) | 108 | 107.0±11.3 | 119.4±12.0 | 123.4±8.7 | 110.3±12.2 | 3.3±1.6 |
| Group C (high KFA) | 48 | 132.8±5.4 | 134.5±7.4 | 130.6±9.4 | 122.5±6.8 | -10.3±1.3 |

**Abbreviation:** KFA, knee flexion angle

knowledge gap regarding the most appropriate timepoint to measure KFA that can best predict, or that most closely correlates with the final post-TKA KFA measurement. Since anesthetic procedures can reduce pain-related limitations, and they can relax the muscles around the knee, the KFA may increase after the administration of anesthesia. However, there is no evidence that correlates the KFA measured under anesthesia with the final KFA. Moreover, the intraoperative drop leg test, which is used by many orthopedic surgeons as a predictor of the final KFA, has not been compared with either the pre-anesthetic or post-anesthetic KFA. To the best of our knowledge, this is the first study to investigate for correlation between the KFA measured at three perioperative timepoints (pre-anesthesia, post-anesthesia, and intraoperatively via the leg drop test) with the KFA at six months post-operation. Although some evidence suggested the plateau phase of recovery in knee motion can improve well past one year and even last up to 2 years [27, 28], our study focuses on up to 6 months of follow-up. This was because the recovery after 6 months usually involves only a slight improvement compared to the magnitude of recovery during the first six months of follow-up [12].

Our findings highlighted that high KFA would be expected while the patients were anesthetized. This means both the intraoperative and immediate postoperative measurements just represent an adequate flexion gap at the time of surgery. If the KFA is diminished at this stage, then the flexion gap is too tight and will not likely recover. We, therefore, recommend using the pre-anesthetic KFA as the predictor of KFA at six months after TKA. The pre-anesthetic KFA in each patient can be applied for individual counseling; however, patients should be openly and clearly informed that there is no way to predict with 100% accuracy that their postoperative KFA will be after TKA. Similarly, Ritter, et al. [25] reported that the pre-anesthetic ROM measurement could best predict the post-TKA ROM. In contrast, Lee, et al. [10] reported that the intraoperative drop leg test KFA could best predict the final ROM after TKA. The difference in finding between our study and the Lee, et al. study may be attributable to differences in study methods and/or surgical techniques. Lee, et al. operated using the cruciate-retained design, and also used a different pain control protocol. They also immediately applied the postoperative continuous passive motion machine, which was not used in our study.

In subgroup analysis, we found only a small number of patients with poor KFA (<90 degrees). Pua Y-H, et al. [29] also reported an incidence of low KFA of approximately 5.6%. However, even though there were only a few patients in this subgroup, we still observed an increase in the KFA at six months compared to the KFA measured preoperatively. It should be noted, however, that the immediately aforementioned improvement was not statistically significant (likely due to the small number of patients), so patient counseling in this group of patients should include this information with some degree of caution. Further study in an adequate number of low KFA patients is needed to confirm the results of our study.

The results of the present study urge us to recommend that surgeons not use the preoperative KFA to predict the final KFA in the high KFA (>120 degrees) subgroup. This is because we observed a decrease in the mean KFA at six months post-operation compared to the preoperative KFA in this subgroup. Our findings are concordant with those of Hirakana et al. and Pasquier G, et al., which showed that a preoperative knee with a smaller flexion angle could gain more flexion angle postoperatively, while a preoperative knee with a larger flexion angle likely loses flexion angle [22, 30].

In fact, the impact of KFA on patient satisfaction differs across countries and continents. In Europe, Thomsen MG, et al. [31] reported that KFA after TKA does not influence patient-perceived outcomes and satisfaction. That finding conflicts with the findings in Asian and Muslim populations [32] whose specific cultural and lifestyle characteristics relative to normal daily activities may influence their compliance with rehabilitation protocol. Kim, et al. [33] conducted a study in Korean TKA and found that patients rated high knee flexion activities as

necessary, and that kneeling difficulty was associated with significant dissatisfaction after TKA. Future study in the effect of the use of the preoperative KFA to counsel patients relative to their predicted postoperative KFA on the level of postoperative patient satisfaction is warranted.

The strengths of this study include it prospective cohort design, and the fact that we evaluated for associations between different perioperative KFA measurement timepoints and the 6-month KFA follow-up. We also analyzed our measurement data according to KFA subgroup [poor (<90 degrees), average (90–120 degrees), and high (>120 degrees)]. The last notable strength of our study is the fact that we defined preoperative KFA as the measurement taken during the perioperative period and just before the administration of anesthesia.

## Limitations

Our study also has a few limitations. First, we only included patients with primary knee osteoarthritis and excluded patients who required MUA after TKA. Therefore, our results might not be generalizable to patients with secondary osteoarthritis, surgical issues related to stiffness, and abnormal biological responses causing arthrofibrosis. Second, we excluded patients who required patellar resurfacing, which may be a preferred technique among some surgeons. Third, there were only nine patients in the poor KFA group, which may be too small of a sample to detect statistically significant differences and associations. Lastly, we did not analyze inter and intra-rater reliability. Nonetheless, we did attempt to minimize the measurement error of the KFA, and we believe that the measurement technique we applied in this study had been proven in the literature with detailed steps to follow.

## Conclusion

Pre-anesthetic KFA had the highest correlation with the final KFA at six months after unilateral primary TKA, especially in the patients who had a preoperative KFA within 90–120 degrees.

## Acknowledgments

The authors gratefully acknowledge the patients who generously agreed to participate in this study and Ms. Nichakorn Khomawut for their data collection assistance. The authors also appreciate Mr. Suthipol Udompunturak, MSc (Applied Statistics), for his statistical analysis. We thank Onlak Ruangsomboon, MD and Kevin P. Jones for their assistance in editing this manuscript.

## Author Contributions

**Conceptualization:** Pakpoom Ruangsomboon, Rapeepat Narkbunnam.

**Data curation:** Polasan Santanapipatkul, Sorarid Sarirasririd.

**Formal analysis:** Pakpoom Ruangsomboon, Chaturong Pornrattanamaneewong.

**Methodology:** Pakpoom Ruangsomboon, Chaturong Pornrattanamaneewong.

**Supervision:** Keerati Chareancholvanich.

**Validation:** Polasan Santanapipatkul.

**Visualization:** Keerati Chareancholvanich.

**Writing – original draft:** Pakpoom Ruangsomboon.

**Writing – review & editing:** Pakpoom Ruangsomboon, Rapeepat Narkbunnam.

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
