## [Decision Letter · Decision Letter 0]

26 Apr 2022

PONE-D-21-16618The pre-anesthetic period is the best time to evaluate the knee flexion angle for predicting the flexion angle after total knee arthroplasty: A prospective cohort studyPLOS ONE

Dear Dr. Narkbunnam,

Thank you for submitting your manuscript to PLOS ONE. After careful consideration, we feel that it has merit but does not fully meet PLOS ONE’s publication criteria as it currently stands. Therefore, we invite you to submit a revised version of the manuscript that addresses the points raised during the review process.

The expert reviewer raises a number of concerns regarding the organization and reporting of the manuscript. In particular, they suggest significant improvements to be made to clarifications of the methodological details and justification of the rationale

Can you please address these concerns in your revision?

We look forward to receiving your revised manuscript.

Kind regards,

Avanti Dey, PhD

Staff Editor

PLOS ONE

Journal Requirements:

Reviewers' comments:

Reviewer's Responses to Questions

**Comments to the Author**

1. Is the manuscript technically sound, and do the data support the conclusions?

Reviewer #1: Partly

2. Has the statistical analysis been performed appropriately and rigorously? 

Reviewer #1: Yes

3. Have the authors made all data underlying the findings in their manuscript fully available?

Reviewer #1: Yes

4. Is the manuscript presented in an intelligible fashion and written in standard English?

Reviewer #1: Yes

5. Review Comments to the Author

Reviewer #1: This prospective study compares the pre anaesthetic, intraoperative and post anaesthetic KFA of patients undergoing TKR with the 6 month KFA concluding that the pre anaesthetic range more closely predicts the 6 month range.

These results are not unexpected and have been reported before with many showing that the final ROM is related to pre operative ROM. This study does add to the literature by delineating the 3 perioperative assessments with the final KFA and so does offer some useful information, however the ROM under anaesthetic shold be higher than pre anaesthesia when pain often limits ROM.

There are some points that need to be clarified:

1 Why were the knees that required MUA excluded from the analysis? I would have expected them to be included.

2 The authors suggest that 6 months evaluation should show the end point improvement in KFA but the cited reference was not indicating that as it was a study of only 25 patients with follow up limited to 6 months. most experienced surgeons would suggest that KFA can improve well past 1 year, even out to 2 years. Studies looking at PROMS show improvement past 1 year which is likely to be related to change in KFA.

3 The authors need to highlight that high KFA would be expected while the patient was anaethetisted and so both the intraoperative and immediate postoperative measurements are really just an indication of creating an adequate flexion gap at the time of surgery. If the KFA is diminished at this stage then the flexion gap is too tight and will never recover.

4 The authors indicate the error with measuring KFA but these are from other studies and the error for this study should be done separately

6. PLOS authors have the option to publish the peer review history of their article (what does this mean?). If published, this will include your full peer review and any attached files.

Reviewer #1: No

---

## [Author Response · Author response to Decision Letter 0]

29 May 2022

Response to reviewers 

May 2022, Plos One, Rapeepat Narkbunnam, Siriraj Hospital, Thailand

Dear Editor and reviewers,

We are very excited to have been given the opportunity to revise our manuscript "The pre-anesthetic period is the best time to evaluate the knee flexion angle for predicting the flexion angle after total knee arthroplasty: A prospective cohort study." We are pleased to submit the revised version for reconsideration for publication in Plos One.

We highly appreciate the detailed and valuable comments from the reviewers. The suggestions are helpful for us, and we shall incorporate most of them in the revised paper. As below, on behalf of my co-authors, I would like to clarify all the points raised by the reviewers, and we hope the reviewers and the Editor will be satisfied with our responses to the comments and the revisions of the original manuscript. 

According to ethical and legal restrictions on sharing a sensitive dataset, the data contain potentially identifying or sensitive patient information; we would like to change the statement of availability of data and material to “Data are available from the Siriraj Institutional Data Access / Ethics Committee (contact via orthoresearch.si@gmail.com) for researchers who meet the criteria for access to confidential data.”

Sincerely yours,

Rapeepat Narkbunnam, M.D.

Associated Professor in Orthopaedics Adult Reconstructive Surgery, 

Department of Orthopaedics Surgery, Siriraj Hospital, Mahidol University, Thailand

Tel: +66(8)81-935-1669 

E-mail address: mai_parma@hotmail.com

---

## [Decision Letter · Decision Letter 1]

24 Aug 2022

PONE-D-21-16618R1The pre-anesthetic period is the best time to evaluate the knee flexion angle for predicting the flexion angle after total knee arthroplasty: A prospective cohort studyPLOS ONE

Dear Dr. Narkbunnam,

Thank you for submitting your manuscript to PLOS ONE. After careful consideration, we feel that it has merit but does not fully meet PLOS ONE’s publication criteria as it currently stands. Therefore, we invite you to submit a revised version of the manuscript that addresses the points raised during the review process.

The manuscript has been evaluated by a further reviewer, and their comments are available below.

The reviewer has raised a number of concerns that need attention. Could you please revise the manuscript to carefully address the concerns raised?

We look forward to receiving your revised manuscript.

Kind regards,

Jamie Royle

Staff Editor

PLOS ONE

Journal Requirements:

Reviewers' comments:

Reviewer's Responses to Questions

**Comments to the Author**

1. If the authors have adequately addressed your comments raised in a previous round of review and you feel that this manuscript is now acceptable for publication, you may indicate that here to bypass the “Comments to the Author” section, enter your conflict of interest statement in the “Confidential to Editor” section, and submit your "Accept" recommendation.

Reviewer #1: All comments have been addressed

Reviewer #2: All comments have been addressed

2. Is the manuscript technically sound, and do the data support the conclusions?

Reviewer #1: (No Response)

Reviewer #2: Partly

3. Has the statistical analysis been performed appropriately and rigorously? 

Reviewer #1: (No Response)

Reviewer #2: Yes

4. Have the authors made all data underlying the findings in their manuscript fully available?

Reviewer #1: (No Response)

Reviewer #2: No

5. Is the manuscript presented in an intelligible fashion and written in standard English?

Reviewer #1: (No Response)

Reviewer #2: Yes

6. Review Comments to the Author

Reviewer #1: (No Response)

Reviewer #2: Thank you for the opportunity to review the manuscript entitled "The pre-anesthetic period is the best time to evaluate the knee flexion angle for predicting the flexion angle after total knee arthroplasty: A prospective cohort study". It is interesting paper and important especially for Asian population. However, this paper needs several modifications before acceptance of publication.

Thank you for the opportunity to review the manuscript entitled "The pre-anesthetic period is the best time to evaluate the knee flexion angle for predicting the flexion angle after total knee arthroplasty: A prospective cohort study". It is an interesting paper and important, especially for the Asian population. However, this paper needs several modifications before acceptance for publication.

General comment

The prediction of the postoperative knee flexion angle is important. It is valuable to know that pre-anaesthesia KFA, not post-anaesthesia or postoperative KFA, is the best correlated with postoperative KFA. However, this study showed only the correlation between the pre-and postoperative KFA. Patients want to know that the preoperative FKA will increase or decrease postoperatively. Therefore, it would be more valuable if the authors show the correlation between preop KFA and change in KFA or absolute KFA value. The negative correlation between preoperative KFA and change in KFA has already been pointed out in the papers.

# Hiranaka T* et al. Is postoperative flexion angle genuinely better in unicompartmental knee arthroplasty than in total knee arthroplasty? A comparison between the knees in the same patients. Knee. 2020 Dec;27(6):1907-1913.

# Kamenaga Tet al. Contralateral knee flexion predicts postoperative knee flexion in unilateral total knee arthroplasty: A retrospective study. Orthop Traumatol Surg Res. 2022 Jan 31:103218.

The authors should analyse the KFA change by referencing this paper. It might also be valuable if the author conducts the regression analysis to predict the absolute value of postop KFA.

Specific comments

Title: As the general comments, regression analysis or correlation between pre-KFA and change in KFA is needed.

Line 108: What is the success rate? The satisfaction rate?

Line 111: To show the decreased ROM (or KFA), the change in ROM (KFA) should be analysed.

Line 138 - 139: How about hip and opposite knee surgery? They must affect the postoperative KFA.

Line 155: "Intraoperative drop leg test KFA. " How was the hip flexion angle? It must largely influence to the knee flexion angle.

Line 161 - 162: "The mechanical concept of prosthesis placement proposed by Insall, et al. [16] was used in all cases." Isn't Insall's technique typical gap balancing technique? However, the author stated that the measured resection technique was used in line 161.

Lines 162 - 164: How did the femoral rotation was decided?

Lines 225 -227: Did these measurements correlate with the KFA?

Line 231: drop leg KFA: Does it mean intraoperative KFA?

Table 3: It is recommendable to analyse the change in KFA.

Lines 268 - 279: The discussion section should start with "The most important finding of the study was that". Most of these sentences should be stated in the introduction section.

Line 290: the pre-anesthetic KFA had the highest correlation with the 6-month KFA. This is the main finding of this study. But, again, the correlation shows the relationship but does not predict the postoperative flexion angle. I recommend analysing the change in FKA or regression.

Line 310 - 313: Please refer to the abovementioned papers.

7. PLOS authors have the option to publish the peer review history of their article (what does this mean?). If published, this will include your full peer review and any attached files.

Reviewer #1: No

Reviewer #2: No

---

## [Author Response · Author response to Decision Letter 1]

30 Sep 2022

Reviewer #1: The prediction of the postoperative knee flexion angle is important. It is valuable to know that pre-anaesthesia KFA, not post-anaesthesia or postoperative KFA, is the best correlated with postoperative KFA. However, this study showed only the correlation between the pre-and postoperative KFA. Patients want to know that the preoperative KFA will increase or decrease postoperatively. Therefore, it would be more valuable if the authors show the correlation between preop KFA and change in KFA or absolute KFA value. 

The negative correlation between preoperative KFA and change in KFA has already been pointed out in the papers.

# Hiranaka T* et al. Is postoperative flexion angle genuinely better in unicompartmental knee arthroplasty than in total knee arthroplasty? A comparison between the knees in the same patients. Knee. 2020 Dec;27(6):1907-1913.

# Kamenaga Tet al. Contralateral knee flexion predicts postoperative knee flexion in unilateral total knee arthroplasty: A retrospective study. Orthop Traumatol Surg Res. 2022 Jan 31:103218.

The authors should analyse the KFA change by referencing this paper. It might also be valuable if the author conducts the regression analysis to predict the absolute value of postop KFA. 

Response to Reviewer #1:

We are pleased and appreciative that you believe our work contributes to scientific knowledge. We appreciate your time and effort in helping us improve the paper. The following are our responses to your suggestions.

RESPONSE: Based on suggested references, to predict the absolute value of 6-month post-operative KFA, we performed linear regression analysis. We agree that showing the direction of correlation will be informative for both the patients and the surgeons. We chose to employ linear regression because it emphasizes both the direction and the quantitative degree of association, thereby being even more informative. Consequently, we analyzed and presented an equation to predict 6-month KFA based on pre-anesthetic KFA for all patients regardless of their baseline KFA, which may be useful in pre-operative counselling. Two line have been added to statistical analysis session mentioned on the regression analysis. (Line 226-227, We employed linear regression analysis to obtain an equation for predicting the final KFA based on the pre-operative KFA.) We added two more line in the result section of the abstract. And we also added three more lines in the result section of full manuscript. (Line 74-76, Predicted 6-month KFA (degrees) adjusted for pre-anesthetic KFA is 45.378 + [0.596 x pre-anesthetic KFA (degrees)] (r= 0.67, p <0.05).)

And, line 261-263, “A formula for predicting 6-month KFA (degrees) from pre-anesthetic KFA is 45.378 + [0.596 x pre-anesthetic KFA (degrees)] in overall patient 

(r= 0.67, p<0.05).”

 We also elaborated our finding from regression analysis and referred to two valuable references that the reviewer kindly suggested in the discussion. Three-line number 286-288 has been added “As a result, attempts have been made to create a system that can accurately predict postoperative KFA in TKA patients. This field of research had been done in a variety of ways, including focusing on the contralateral KFA to predict the final KFA [15].” 

And, line 330-333 “Our findings support the same results pattern as a study from Hirakana et al., and Pasquier G, et al., which showed that a preoperative knee with a smaller flexion angle gains more flexion postoperatively, but a preoperative knee with a bigger flexion angle loses flexion angle. [22,30]”

Specific comments

Title: As the general comments, regression analysis or correlation between pre-KFA and change in KFA is needed.

Response: We analyzed in accordance with your recommendation, as previously described.

Line 108: What is the success rate? The satisfaction rate?

Response: We apologies for the ambiguity that our prior wording may have caused. We revised the term “success rate” to “survival rate”. Line 109-110 have been revised to “Total knee arthroplasty (TKA) has been a successful treatment for end-stage knee osteoarthritis for at least 25 years, with a reported 82% survival rate.”

Line 111: To show the decreased ROM (or KFA), the change in ROM (KFA) should be analysed.

Response: As previously indicated, we conducted an analysis in accordance with your recommendation and presented the change in KFA in Table 3.

Line 138 - 139: How about hip and opposite knee surgery? They must affect the postoperative KFA.

Response: We sincerely apologize for the misunderstanding we may have caused by our earlier usage of a phrase that was not as well-defined. The authors strongly agreed that these two circumstances suggested by reviewers could have an impact on the KFA, and we have excluded them from our recruitment process. By excluding patients with “previous surgery”, we already excluded those with all previous hip and knee replacement in our screening process. In the revised manuscript, we attempted to better clarify the exclusion criteria. (Line 139-140: “The exclusion criteria were 1) Prior hip and knee replacement surgery on any side”). We added rationales and reference in line 144-145 (“Because prior hip and opposite knee surgery can alter the post-operative KFA, these conditions were excluded. [15]”). Moreover, we also elaborated on this condition in our discussion part (Line 286-288: “As a result, attempts have been made to create a system that can accurately predict postoperative KFA in TKA patients. This field of research had been done in a variety of ways, including focusing on the contralateral KFA to predict the final KFA [15].) 

Line 155: "Intraoperative drop leg test KFA. " How was the hip flexion angle? It must largely influence to the knee flexion angle.

Response: The hip flexion angle was perpendicular to the operative table. According to Lee et al. [10] who purposed the drop leg test, after capsular closure, intraoperative flexion against gravity or drop leg test was acquired by passively flexing the patient's hip 90 degrees and enabling the patient's lower leg weight to flex the knee joint. Previous figure 1 may not well illustrate the drop leg test as intended. Accordingly, we revised the attached figure1 as attached hereby. “Line 156-159 Figure Captions

Figure 1. Knee flexion angle (KFA) measurement using a long protractor double-arm standard goniometer (left). Intraoperative drop leg test KFA was measured using a long arm sterile protractor after placement of the prosthesis and capsular closure (right).”

We also included drop leg test details to the materials and methods section to provide the reader with clearer information. (Line 171-173 Intraoperative flexion against gravity or intraoperative drop leg test was conducted after capsular closure by passively flexing the patient's hip 90 degrees and allowing the patient's lower leg weight to flex the knee joint [10].)

Line 161 - 162: "The mechanical concept of prosthesis placement proposed by Insall, et al. [16] was used in all cases." Isn't Insall's technique typical gap balancing technique? However, the author stated that the measured resection technique was used in line 161.

Response: Our prior definition may have led to misunderstanding. We recognized that the technique described by Insall et al. in CORR 1985 utilized gap balancing, but the alignment was perpendicular to the mechanical axis. As previously described by Insall, we sought for the prosthesis alignment to be perpendicular to the mechanical axis when performing TKA. To eliminate the previous ambiguity, we have altered our explanation of the procedure (Line 164-166: In all cases, surgeons targeted femoral and tibial cuts perpendicular to the mechanical axis according to the preoperative angles as measured by preoperative scanogram. [17])

Lines 166 – 170: How did the femoral rotation was decided?

Response: Thank you for reminding us this important point. Line 166-170 have been added “The rotation of the femur was adjusted to achieve a balanced, rectangular flexion gap and proper patellar tracking. To obtain a rectangular flexion gap, we set femoral rotation parallel to or with slightly external rotation to the surgical transepicondylar axis.”

Lines 225 -227: Did these measurements correlate with the KFA?

Response: Thank you for bringing up these interesting issues. In table 1, we attempted to demonstrate, given demographic details, that there were no extreme outliers among our cohort demographics, for example, no morbid obesity patients and no dwarfism or acromegaly patients recruited in our study. Although significant correlations can be seen if all these characteristics were analyzed for Pearson’s correlation coefficients, these results can also occur by chance due to multiple analyses and do not represent true associations. Therefore, we strongly believe that a strong scientific rationale should be the basis of our decision to undertake correlation analyses between specific patient characteristics and KFA rather than analyzing all collected demographics. According to the earlier studies by Nassif et al., Brian et al., Dere et al., and Farahini et al., these patient demographic variables had no significant effect on KFA. Furthermore, our cohort's gender, age, BMI, and height details were not significantly skewed to extreme values that could influence post-operative KFA.

 However, lines 236-237 were added to highlight this result: "There were no extreme outliers over 2.5 SD in our cohort's demographic data.".

Line 231: drop leg KFA: Does it mean intraoperative KFA?

Response: Sorry for the confusion these ambiguous terms might have caused. We have revised phrase from “drop leg KFA” to “the intraoperative drop leg KFA” at line 175, 241 and Table 3 sub header (line 275).

Table 3: It is recommendable to analyzed the change in KFA.

Response: We carried out an investigation in accordance with your suggestion, as was previously declared, and the change in KFA is now presented in Table 3.

 

Lines 268 - 279: The discussion section should start with "The most important finding of the study was that". Most of these sentences should be stated in the introduction section.

Response: We have rearranged the first part of our discussion as per your suggestion. Line 280-288 have been revised “The most important finding of the study was that the correlation between the pre-anesthetic KFA and the 6-month KFA was the highest. The post-anesthetic KFA and intraoperative drop leg test KFA might not be appropriate predictors of 6-month KFA because they yielded insignificant and negligible correlations. This is an important finding because KFA is utilized in clinical practice to inform patients about their predicted postoperative KFA. If the given KFA prediction is inaccurate, TKA patients may be dissatisfied if their postoperative KFA does not meet their expectations. As a result, attempts have been made to create a system that can accurately predict postoperative KFA in TKA patients. This field of research had been done in a variety of ways, including focusing on the contralateral KFA to predict the final KFA [15].”

Line 290: the pre-anesthetic KFA had the highest correlation with the 6-month KFA. This is the main finding of this study. But, again, the correlation shows the relationship but does not predict the postoperative flexion angle. I recommend analysing the change in FKA or regression.

Response: Thank you for your great suggestion. As stated previously, we conducted further analyses and reported them in accordance with your recommendation.

Line 310 - 313: Please refer to the abovementioned papers.

Response: As previously stated, we have cited the aforementioned articles in the method section, discussion section, and added references.

---

## [Decision Letter · Decision Letter 2]

19 Jan 2023

The pre-anesthetic period is the best time to evaluate the knee flexion angle for predicting the flexion angle after total knee arthroplasty: A prospective cohort study

PONE-D-21-16618R2

Dear Dr. Narkbunnam,

We’re pleased to inform you that your manuscript has been judged scientifically suitable for publication and will be formally accepted for publication once it meets all outstanding technical requirements.

Kind regards,

Carlos M. Isales, M.D.

Academic Editor

PLOS ONE

Additional Editor Comments (optional):

Reviewers' comments:

Reviewer's Responses to Questions

**Comments to the Author**

1. If the authors have adequately addressed your comments raised in a previous round of review and you feel that this manuscript is now acceptable for publication, you may indicate that here to bypass the “Comments to the Author” section, enter your conflict of interest statement in the “Confidential to Editor” section, and submit your "Accept" recommendation.

Reviewer #1: All comments have been addressed

Reviewer #2: All comments have been addressed

2. Is the manuscript technically sound, and do the data support the conclusions?

Reviewer #1: Yes

Reviewer #2: Yes

3. Has the statistical analysis been performed appropriately and rigorously? 

Reviewer #1: Yes

Reviewer #2: Yes

4. Have the authors made all data underlying the findings in their manuscript fully available?

Reviewer #1: Yes

Reviewer #2: Yes

5. Is the manuscript presented in an intelligible fashion and written in standard English?

Reviewer #1: Yes

Reviewer #2: Yes

6. Review Comments to the Author

Reviewer #1: (No Response)

Reviewer #2: Thank you for the great effort of the authors. I think the manuscript has been well revised and is suitable for publication in Plos one.

7. PLOS authors have the option to publish the peer review history of their article (what does this mean?). If published, this will include your full peer review and any attached files.

Reviewer #1: No

Reviewer #2: No

---

## [Editor Report · Acceptance letter]

26 Jan 2023

PONE-D-21-16618R2 

The pre-anesthetic period is the best time to evaluate the knee flexion angle for predicting the flexion angle after total knee arthroplasty: A prospective cohort study 

Dear Dr. Narkbunnam:

I'm pleased to inform you that your manuscript has been deemed suitable for publication in PLOS ONE. Congratulations! Your manuscript is now with our production department. 

Kind regards, 

on behalf of

Professor Carlos M. Isales 

Academic Editor

PLOS ONE